# OpenReview forum: "FactTest: Factuality Testing in Large Language Models with Statistical Guarantees"
_ICLR.cc/2025/Conference — Submitted to ICLR 2025_

### Official Review · Reviewer_sEhD · 2024-10-19

**Soundness:** 3
**Presentation:** 3
**Contribution:** 3
**Rating:** 8
**Confidence:** 3

**Summary:**

This paper offers a hypothesis testing framework to control the Type I error, while also showing that the Type II remains sufficiently low. The results hold for general binary classification tasks, and they are related to standard conformal prediction and calibration results. The authors frame their work in the context of LLM hallucination detection, and employ relevant scoring functions from the literature for this task. While the theory holds in distribution, they also provide an extension to out-of-distribution via density ratio estimation.

The methodology is justified by theoretical results. In practice, the method seems relatively simple to use, and it appears to provide notable improvements over baselines.

**Strengths:**

The paper is well written and theoretically justified. Experiments seem to support the theoretical claims.

**Weaknesses:**

In the conformal prediction literature, density ratio estimation is a fairly standard way to extend in-distribution guarantees to out-of-distribution. Yet, these estimates tend to be unreliable. Would be interested for the authors to comment on the robustness of their method with respect to the density ratio estimates.

**Questions:**

- Is there a typo in the legend of Figure 1? The ve10 and ve15 results are repeated twice.
- I feel the authors could bring a stronger connection with the calibration and conformal prediction literature. The UQ of LLMs paragraph in the work related section may mention previous work on calibration of confidence scores in LMMs.
- The threshold selection approach used in the paper essentially corresponds to a conformalized quantile regression. This helps achieving a marginal (on average over all the data) guarantee on the Type I error. However, as the error may not be homogeneous over different questions, one may wonder if ensuring a marginal guarantee is, in fact, sufficient. I wonder if the authors have experienced different error magnitudes across different segments of the datasets.

---

> ### Author Response · Authors · 2024-11-21
> **Response to Reviewer sEhD (1)**
>
> Thank you for the positive feedback and constructive suggestions that can help us to improve the paper. We are happy that you acknowledge our work’s originality, technical quality and presentation. For your questions, we provide additional experimental results and explanations as follows (also see the revised paper in the updated pdf):
>
> > ***W1: Robustness with respect to density ratio estimates***
>
> **R1:** Thank you for your comment. Indeed, density ratio estimation can be unreliable in some cases, such as in high dimensional spaces. In theory, our method will be affected by the density ratio estimation error.
>
> In the covariate shift setting, the source conditional distribution $\tilde P_{y|q,M(q)}$ equals the target conditional distribution $P_{y|q,M(q)}$. If we assume the oracle score function $\eta(q,M(q))=ℙ_{y\sim P_{y|q,M(q)}}(y=1|q,M(q))$ can be estimated well and we have access to unlabeled samples from the target distribution $P_{q,M(q)}$, one way for decreasing the impact of density ratio estimation is using semi-parametric theory to construct a debiased classifier, which is doubly robust with respect to the errors of density ratio estimator and oracle score estimator. Then the error of density ratio estimator will appear through a multiplication with the error of oracle score estimator.
>
> However, in our cases, the oracle score function corresponds to the oracle rule of judge whether an answer is correct for a question, which is itself a challenging and unsolved problem. Therefore, it is not clear how to construct a good estimator for the oracle score function. Consequently, for hallucination detection of LLMs, it is hard to bypass the dependence on the first order density ratio estimation error.
>
> > ***Q1: Is there a typo in the legend of Figure 1? The ve10 and ve15 results are repeated twice.***
>
> **R1:** Thank you for pointing out the problem and sorry for the confusion caused by our typos. The legends should be ve10, ve15, se10 and se15. We have corrected this in the revised pdf.
>
> > ***Q2: I feel the authors could bring a stronger connection with the calibration and conformal prediction literature.***
>
> **R2:** Thank you for your advice. We have added the literature review of calibration an conformal prediction in the revised paper (See Sec 5 and B). Here we briefly summarize these works:
>
> **More related works about Confidence Calibration:** Recent research on confidence calibration for LLMs has explored several innovative approaches. For example, Tian et al. (2023) elicits verbalized confidences to improve calibration. Huang et al. (2024) proposes confidence elicitation methods for long-form generations. Multicalibration (Detommaso et al., 2024) aims to ensure LLM confidence scores accurately reflect the true likelihood of predictions being correct.
>
> **More related works about Conformal Prediction:** Conformal prediction is a statistical framework that provides finite-sample, distribution-free guarantees on the uncertainty of predictions. It enables the construction of confidence sets that contain the true outcome with a specified probability (Shafer & Vovk, 2007; Angelopoulos & Bates, 2022; Barber et al., 2023). Specifically, Kumar et al. (2023) provides conformal guarantees on multiple-choice datasets. C-RAG (Kang et al., 2024) provides conformal risk analysis for RAG models and certifies an upper confidence bound. CLM (Quach et al., 2024) extends conformal prediction for open-form generations and provide coverage guarantees. Conformal Factuality (Mohri & Hashimoto, 2024) enables the application of conformal prediction in improving model performance. FactTest differs from those works in that it aims to evaluate the model's ability to answer correctly and abstain from answering unknown questions.

---

> ### Author Response · Authors · 2024-11-21
> **Response to Reviewer sEhD (2)**
>
> > ***Q3: The threshold selection approach used in the paper essentially corresponds to a conformalized quantile regression. This helps achieving a marginal (on average over all the data) guarantee on the Type I error. However, as the error may not be homogeneous over different questions, one may wonder if ensuring a marginal guarantee is, in fact, sufficient. I wonder if the authors have experienced different error magnitudes across different segments of the datasets.***
>
> **R3:** Thank you for your insightful question.
>
> Although the error may not be homogeneous across different answers and domains, ensuring a marginal guarantee is suficient for Type I error control over the entire test distribution. However, we acknowledge that this marginal guarantee on the Type I error may not fully capture variations in error rates across different segments of the data, especially if the distributions over different domains are quite different. We will leave this conditional Type I error control as an important future work, by leveraging the tools developed in recent literature on the conditional coverage of conformal prediction.
>
> In the following, we provide experiment results of FactTest-kle15 on ParaRel using Llama 13B across different segments of the dataset. Since the testing dataset contains 15 different subsets according to the domains of the questions, we report the corresponding accuracy and Type I error of each domain. The significance level is set to 0.05. 'All' includes all the subsets.
>
> | metric\subset | All    | field of work | occupation | employer | genre | native language | capital of | named after | religion | headquarters location | manufacturer | developer | place of birth | twinned administrative body | place of death | record label |
> | ------------- | --- | ------------- | ---------- | -------- | ----- | --------------- | ---------- | ----------- | -------- | --------------------- | ------------ | --------- | -------------- | --------------------------- | -------------- | ------------ |
> | Accuracy      |  78.45   | 5.88          | 44.44      | 86.11    | 15.62 | 97.81           | 59.25      | 80.00       | 54.54    | 94.18                 | 98.99        | 95.83     | 85.71          | 0                           | 0              | 0            |
> | Type I error  |  0.03   | 0.05          | 0.02       | 0.02     | 0.06  | 0.10            | 0.04       | 0.02        | 0.03     | 0.03                  | 0.02         | 0.01      | 0.01           | 0.01                        | 0.00           | 0.03         |
>
> As shown in the table, while the overall Type I error across the entire dataset is controlled at the specified significance level (0.05), there is variability in the Type I error across different domains.

---

> > ### Comment · Reviewer_sEhD · 2024-12-02
> >
> > Thank you for your thorough answers, appreciated. I will keep my score.

---

> > > ### Author Response · Authors · 2024-12-03
> > >
> > > Thank you for your response and your suggestions to make our work better.

---

### Official Review · Reviewer_roCe · 2024-11-03

**Soundness:** 2
**Presentation:** 3
**Contribution:** 1
**Rating:** 3
**Confidence:** 4

**Summary:**

This paper proposes a novel statistical method that detects factuality of large language models (LLMs) with statistical guarantees, i.e. hallucination detection with guarantees. The main method assumes the iid assumption, and controls the type 1 error by thresholding uncertainty of LLM’s answers. This is further extended to hold in covariate shift by using rejection sampling. The efficacy of methods is validated over question-answering and multiple-choice tasks with white-box and black-box models.

**Strengths:**

I like this paper as mainly this tackles an important and timely problem.

* This paper attacks an important problem, combatting hallucination in LLMs with guarantees.
* The paper is well-written and easy to follow.
* This paper has Fairly extensive experiments.

**Weaknesses:**

This paper is quite interesting, but I am mainly leaning to rejection due to the novelty of this paper – this paper completely ignores existing papers in conformal prediction and selective prediction, which are popular methods for building trustworthy AI models in general.

* (1) and (3) are equivalent to PAC-style conformal prediction (See proposition 2b in https://arxiv.org/abs/1209.2673 or other related papers). What is the novelty of the proposed method with respect to the PAC-style conformal prediction?
* In language tasks, obtaining the correctness of answers is the most important issue. Otherwise, we can apply traditional techniques for LLMs (e.g., conformal prediction). Conformal language modeling (https://arxiv.org/abs/2306.10193) proposes to extend conformal prediction for LLMs and this method can be used as a detector by checking whether a generated answer is included in a conformal set. What’s the novelty of the proposed method with respect to this conformal language modeling? How can you obtain the indicator variable y_i in Section 2.2?
* To my understanding, (4) is incorrect. This is barely an application of a binomial tail bound under the iid assumption, but as written in the paper it is not iid, i.e., the outer probability is taken over D but the inner probability is taken over a filtered distribution P_0. Can you justify the correctness of (4) if I miss something?
* If this paper’s method is equivalent to PAC conformal prediction, extension to covariate shift via rejection sampling is not novel. In particular, https://arxiv.org/abs/2106.09848 extends the PAC conformal prediction under covariate shift using the same rejection sampling techniques. What’s the novel part of the proposed method compared to this existing work?
* In experiments, an important baseline is missing. Can you compare your method and PAC conformal prediction in experiments for both no shift and covariate shift cases?

**Questions:**

Previously mentioned Weaknesses end with questions. Please refer to those questions.

---

> ### Author Response · Authors · 2024-11-21
> **Response to Reviewer roCe (1)**
>
> Thank you for the questions you raise to help improve our paper. We are happy that you acknowledge our work’s motivation, presentation and experiments. For your questions, we provide more explanations and additional experimental results to address your concerns (also see the revised paper in the updated pdf):
>
> > ***Q1: (1) and (3) are equivalent to PAC-style conformal prediction. What is the novelty of the proposed method with respect to the PAC-style conformal prediction?***
>
> **R1:** Thank you for pointing out the reference Vovk (2012)[1] of PAC-style conformal prediction. We would like to clarify the difference between our method and the PAC-style conformal prediction:
> 1) Conformal prediction aims to produce a prediction set for the correct output, but we aim to test whether the output is incorrect or not.
> 2) Conformal prediction typically treats all samples equally, however, we treat the correct and incorrect samples differently.
> 3) The power analysis for conformal prediction mainly focuses on the sizes of prediction sets, but we study the type II error (misclassifying correct answers as incorrect) of our methods.
>
> Our novelty can be summarized as follows:
> 1) We are the first to formulate hallucination detection as a hypothesis tesing problem.
> 2) Motivated by Neyman-Pearson classification, instead of constructing prediction sets for the correct answers using conformal prediction, we propose an one-sided hypothesis testing for the incorrectness of answers. Unlike conformal prediction where all samples are used equality, we prioritize detecting incorrect answers and utilize only incorrect samples in the calibration data.
> 3) We study the type II error of our method, while the power analysis for conformal prediction are mainly about the sizes of prediction sets.
>
>
> > ***Q2: What’s the novelty of the proposed method with respect to conformal language modeling? How can you obtain the indicator variable y_i in Section 2.2?***
>
> **R2:** Thank you for your question. Our work relates to conformal language modeling, and we have included it in Sec. 5 in updated paper. However, our work is different from it particularly in the following aspects:
>
> 1) The goals of our method and conformal language modeling[2] is different. Our goal is to detect incorrect answers, while conformal language modeling aims to generate correct answers.
> 2) Since the goals are different, the outputs of these two frameworks are also different. Suppose we ask the language model $M$ a hard question $q$, such that $M$ is likely to generate incorrect answers. In this case, if our algorithm thinks the answer is indeed incorrect, we replace its answer by "I don't know" and terminate. However, what conformal language modeling does is to ask $M$ to keep generating (likely incorrect) answers, until it thinks there exists a correct answer, and then output a prediction set containing multiple answers.
> 3) CLM is proposed to provide coverage guarantees while our framework provide guarantees on Type I error.
> 4) In addition to type I error control, our work are also guaranteed to have small type II error. To the best of the authors' knowledge, conformal language modeling doesn't have such power analysis.
>
> Besides, we have to state that "using CLM as a detecter by checking whether a generated answer is included in a conformal set" is not feasible. The prediction set only guarantees that there exists a correct answer in the conformal set with high probability, with no guarantees on including all possible correct answers or including no incorrect answers. It is highly likely that a correct answer is not included in their prediction set or an incorrect answer is included in their prediction set. If we only reject answers not in the conformal set, all the incorrect answers inside the conformal set will be accepted, leading to a large type I error. Meanwhile, all the correct answers not in the conformal set will be rejected, resulting in a large type II error. Moreover, if one increases the coverage of the conformal set, this more reliable conformal set will lead to a larger type I error, since more incorrect answers will be included in the conformal set.
>
> As for the indicator variable $y_i$, we use greedy decoding to get the realization of $M(q)$ and then it depends on task. Following R-Tuning[3], for multiple-choice datasets, $y_i = \mathbf{1}[M(q_i)=a_i]$. For short-form question-answering datasets, where the ground truths are typically numbers, words or phrases, we set $y_i = \mathbf{1}[a_i \subseteq M(q_i)]$, which means a generated answer is considered correct only if it contains the provided answer. We have added these details in Sec 4.1 in updated PDF.

---

> ### Author Response · Authors · 2024-11-21
> **Response to Reviewer roCe (2)**
>
> > ***Q3:  (4) is barely an application of a binomial tail bound under the iid assumption, but as written in the paper it is not iid, i.e., the outer probability is taken over D but the inner probability is taken over a filtered distribution P_0. Can you justify the correctness of (4) if I miss something?***
>
> **R3:** Thank you for your question. Equation (4) in our paper is correct. Since our algorithm only utilize the incorrect samples, although we count the randomness of all samples in the outer probability, the event in the probability only depends on incorrect samples, therefore the i.i.d. condition is still valid.
>
> > ***Q4:  If this paper’s method is equivalent to PAC conformal prediction, extension to covariate shift via rejection sampling is not novel.***
>
> **R4:** As stated in Questions 1 and 2, both our target and method are not equivalent to PAC conformal prediction. Indeed, rejection sampling is a standard method for distribution shift. We are actually adopting this standard technique to the new problem of hallucination detection and adapt our framework to out-of-distribution settings.
>
> > ***Q5:  Can you compare your method and PAC conformal prediction in experiments for both no shift and covariate shift cases?***
>
> **R5:** Thank you for your suggestion. However, we have to stated that the goal of PAC conformal prediction is different from ours. Besides, traditional conformal prediction, including PAC-style ones, is not suitable for the generation of LLMs since the output space is infinite and it's infeasible to explore all possible predictions.
>
> As you mentioned before, Conformal Language Modeling (CLM) extends traditional conformal prediction to language generation by calibrating a stopping rule and employing a rejection rule, which is also PAC-style.
>
> Though we have stated above that checking whether an output is from CLM prediction set isn't a suitable method for hallucination detection, we still compare FactTest with Conformal Language Modeling (CLM) on ParaRel and HotpotQA to address your concern.
>
> Table: The accuracy of FactTest-kle15 and CLM.
>
> | Dataset  | Model        | Pretrained | CLM   | FactTest-kle15 |
> | -------- | ------------ | ---------- | ----- | -------------- |
> | ParaRel  | Openllama-3B | 36.66      | 39.86 | 78.45          |
> |          | Openllama-7B |  40.38     | 42.58 | 76.83               |
> | HotpotQA | Openllama-3B |  25.72     | 26.40  |           55.35     |
> |          | Openllama-7B |  28.63     | 30.82  |           60.66     |
>
>
>
> **References**
>
> [1] Conditional validity of inductive conformal predictors, PMLR 2012.
>
> [2] Conformal Language Modeling, ICLR 2024.
>
> [3] R-Tuning: Instructing Large Language Models to Say 'I Don't Know', NAACL 2024.

---

> ### Author Response · Authors · 2024-11-24
> **We would like to hear back from reviewer roCe**
>
> Dear reviewer roCe,
>
> We would like to follow up to see if the responses address your concerns or if you have any further questions. We would really appreciate the opportunity to discuss this further if our response has not already addressed your concerns. Thank you again!

---

> ### Comment · Reviewer_roCe · 2024-11-26
>
> Thanks for the author’s well-organized answers. I would still not be convinced of the novelty of this paper so maintain my score mainly due to the following reasons.
> * Achieving (3) is exactly the goal of conformal prediction except that this paper only considers incorrect samples – I think considering only incorrect samples can be a  difference but incremental. The paper would be more convincing if it introduces the core of conformal prediction and then highlights the differences.
> * The CLM is the main competitor of this paper (as the goals of this paper and the conformal prediction are aligned) but authors claim that it is not true. In particular, detecting the hallucination by checking the membership of a generated answer from a conformal set is possible by exploiting an admissible function (in CLM) or the method obtaining y_i (in this paper) – I feel that authors ignore in-depth relation between their method and the CLM/conformal prediction but checking superficial differences for claiming novelties.
> * The details on the comparison with the CLM are missing, so I cannot judge the correctness results. At least, authors should use indicator loss instead of a general loss in the CLM for a right comparison.
>
> Given very extensive experiments, addressing the above and acknowledging existing works would clearly enhance the reliability of the paper.

---

> ### Author Response · Authors · 2024-11-27
> **Response to Reviewer roCe**
>
> > Achieving (3) is exactly the goal of conformal prediction except that this paper only considers incorrect samples
>
> Thank you for your follow-up comment. We would like to clarify how our work distinguishes itself from existing conformal prediction (CP) methods in LLMs and to highlight our unique contributions to the field of hallucination detection.
>
> **Our work fundamentally builds upon the Neyman-Pearson (NP) classification framework**[1] to establish a threshold selection mechanism that ensures control over Type I error in the context of hallucination detection. **Although NP classification does not explicitly reference conformal prediction**, seeing the inherent connection, **we have demonstrated that the relationship between conformal prediction and NP classification is akin to the well-known duality between confidence interval and hypothesis testing**, and **provided a formal proposition** to show that **the technique employed in the NP umbrella algorithm is equivalent to that used in PAC-style conformal prediction** [2] for determining membership based on p-values (**See Sec.C Discussion in the update PDF**). By defining a classifier and calibrating solely on incorrect samples, we can effectively reformulate PAC conformal prediction to suit our specific problem of hallucination detection. **This novel approach of defining a plug-in classifier and focusing calibration exclusively on incorrect samples** represents a simple yet significant advancement over traditional methods and offers a new perspective on applying CP to LLMs. Moreover, we identify the optimal score function for constructing the optimal classifier with minimum type II error. This aspect has not yet been explored in the conformal prediction literature.
>
> Overall, our main contributions apart from NP classification and PAC conformal prediction are as follows:
>
> - We **take the first step to formulate hallucination detection as a hypothesis testing problem**, explicitly aiming to control both Type I and Type II errors. Traditional CP methods in LLMs, such as Conformal Language Modeling (CLM), focus primarily on coverage guarantees without differentiating between correct and incorrect samples, thereby **lacking explicit error rate controls** essential for reliable hallucination detection (See our next response for more details).
> - We define **a plug-in classifier to predict the correctness** of question-answer pairs and **utilizes only incorrect samples for calibration**. This targeted calibration allows for precise control over Type I errors by specifically modeling the distribution of incorrect answers, unlike CP’s uniform treatment of all samples.
> - In addition to controlling Type I error, **our framework provides a novel Type II error control analysis, which is not addressed in the NP umbrella algorithm[1] or PAC conformal prediction frameworks**. This dual-error control ensures that incorrect answers are reliably rejected while correct answers are not unnecessarily excluded, thereby enhancing the overall reliability of our factuality testing in LLMs.
> - We perform extensive experiments on question-answering (QA) and multiple-choice benchmarks. The empirical results demonstrate that FactTest is not only **simple to use** but also **highly effective** in detecting hallucinations, achieving 40\% accuracy improvement on ParaRel, WiCE and FEVER, and 30\% on HotpotQA.
>
> [1] Neyman-Pearson Classification Algorithms and NP Receiver Operating Characteristics, Science Advances 2018.
>
> [2] Conditional Validity of Inductive Conformal Predictors, PMLR 2012.

---

> ### Author Response · Authors · 2024-11-27
> **Response to Reviewer roCe**
>
> > The CLM is the main competitor of this paper
>
> While both approaches aim to ensure correctness, there are critical distinctions that render Conformal Language Modeling (CLM) unsuitable for effective factuality testing.
>
> At first glance, the goals of our paper and CLM may appear similar, as both seek to provide correct answers. Furthermore, the admissible function $A$ in CLM could be chosen as the correctness indicator $y$ in our framework. However, a closer examination reveals that CLM is fundamentally inadequate for factuality testing. Specifically, CLM fails to provide guarantees for controlling either type I or type II errors..
>
> Using our notations, CLM constructs a set $\mathcal{C}(q)$ of answers $\tilde a$ for a given question $q$ with property $$\mathbb{P}(\mathbb{P}(\exists \tilde a\in\mathcal{C}(q):A(\tilde a)=1|\mathcal{D})\ge 1-\alpha)\ge 1-\delta.$$
> Roughly speaking, the conformal set $\mathcal{C}(q)$ guarentees to contain at least one correct answer for $q$ with high probability. However, $\mathcal{C}(q)$ has the following problems in factuality testing:
>
> 1) The set $\mathcal{C}(q)$ is not guaranteed to contain only correct answers. On the contrary, to ensure the $1-\alpha$ coverage of $\mathcal{C}(q)$ for a correct answer, it is likely to contain incorrect answers. If we only reject answers outside $\mathcal{C}(q)$, all the incorrect answers in $\mathcal{C}(q)$ will be misclassified as correct. In extreme cases, to ensure a 100\% coverage of $\mathcal{C}(q)$, $\mathcal{C}(q)$ should contain all possible answers, then no answer will be rejected and the type I error becomes 100\%. Therefore, CLM does not control type I error.
>
> 2) CLM guarantees that $C(q)$ contains at least one correct answer but does not account for cases where $q$ has multiple correct answers. Any correct answer not included in $\mathcal{C}(q)$ will be misclassified as incorrect. As a result, CLM provides no control over type II error.
>
> In summary, the approach of detecting hallucinations by verifying whether a generated answer belongs to CLM conformal set is inherently infeasible for effective factuality testing.
>
> > The details on the comparison with the CLM are missing.
>
> Thank you for your suggestion. We provide more details as follows to address your concerns.
>
> To ensure fair comparison, we utilize the indicator $y_i$ from our paper as the admission function in CLM and just as you said, employ the indicator loss. We utilize the Algorithm 1 in CLM to construct conformal set since we do not need to select individual components in our datasets. Specifically, we use the likelihood function of the base LM with length-normalization to serve as $\mathcal{Q}(x,y)$, and MAX as $\mathcal{F}$, consistent with CLM's original setup. We utilize the code provided by CLM to implement these functions. For $k_{max}$, we set it to 20, adhering to CLM's configuration. For $\epsilon$, as mentioned in the paper, not all values of $\epsilon$ are valid. In our experiments, for example, there doesn't exist a valid configuration when $\epsilon<0.4$ on ParaRel, we reported the result with $\epsilon=0.8$. However, the results for other $\epsilon$ can be seen as follows.
>
> Table: The accuracy of FactTest-kle15 and CLM.
> |              | Pretrained | CLM,$\epsilon=0.5$ | CLM,$\epsilon=0.6$ | CLM,$\epsilon=0.7$ | CLM,$\epsilon=0.8$ | CLM,$\epsilon=0.9$ | FactTest-kle15 |
> | ------------ | ---------- | ------------------ | ------------------ | ------------------ | ------------------ | ------------------ | -------------- |
> | OpenLlama-3B | 36.66      | 38.08              | 37.45              | 38.02              | 39.86              | 38.67              | 78.45          |
> | OpenLlama-7B |  40.38    |    43.11     |      42.72              |        42.31           |       42.58             |        55.37            | 76.83          |

---

### Official Review · Reviewer_Fj97 · 2024-11-03

**Soundness:** 1
**Presentation:** 1
**Contribution:** 2
**Rating:** 3
**Confidence:** 3

**Summary:**

The paper proposes a framework, FactTest to reduce hallucinations in LLM responses. FactTest uses Neyman-Pearson methods to make an uncertainty classifier and prevents LLMs from answering responses for which they are uncertain to reduce hallucination. The empirical analysis shows the method's constrained Type-1 errors and accuracy.

**Strengths:**

1. The paper applies the Neyman-Pearson method to construct an uncertainty classifier with guarantees on constrained Type-1 error.
2. The work suggests a method to remove the typical iid assumption when designing Neyman-Pearson classifiers to cater to practical scenarios.
3. The experiments show applicability of the method to black-box API models as well, thus enhancing its practical value.

**Weaknesses:**

I have the following major criticisms for the paper. With these, I think that the paper is not ready for acceptance. However, if the authors can address my concerns appropriately, I can consider increasing my score.
1. My main concern is the assumption of the equivalence of the notions of certainty of the model and its correctness. The method builds entirely on the premise that if the model is certain then it is going to be correct. The wording of the paper conveys this too, where the words "certain" and "correct" are often used interchangeably. Lines 97-104 mix up the notions of correctness and certainty. As the paper itself mentions in line 34 that models can generate incorrect responses with high confidence, such an equivalence of the notions of correctness and certainty is incorrect till certainty is formally defined differently from prior works, which is not done in the paper. I am especially confused by line 102, which says that when the null hypothesis is rejected, i.e., the model can answer q certainly, then M(q) aligns with a. There is no justification given for the same till that point of the paper. Same confusion is created in lines 124-125 where first $y_i$ indicates uncertainty of responses and then correctness in the following equation.
2. There is a mismatch in the definition of hallucination in the Introduction. Line 34 mentions hallucination as the models generating incorrect responses with high confidence and line 42 says that hallucination occurs when model is uncertain. The authors should consistently define the property.
3. I am doubtful about the theoretical generalizability of the uncertainty predictor $\hat{f}_\alpha$, which is constructed on the samples from $\mathcal{D}$ to samples outside of $\mathcal{D}$, to be useful as a general uncertainty calibrator. I believe that the authors should thoroughly study this aspect. Does the sample space of $P$ also contain elements outside of $\mathcal{D}$?
4. "We prove that our statistical framework achieves strong power control under mild conditions, ensuring that the predictor can also maintain a low Type II error." I don't see how this is a contribution from the main paper. The result of using a thresholding based classifier that has controlled Type 2 error appears to follow from Tong (2013). As the authors claim this contribution, they should provide at least a proof sketch for Theorem 1.
5. There are several instances of using terms before definition, some of which I enumerate below.
    1. Lines 50-54 mention terms like Type-1 error, Type-2 error, and false positive rate, before clearly state the null hypothesis.
    2. [Line 62] The term "human-annotated samples" is used before definition/context.
    3. $\epsilon_{\eta}$ is used in line 170 before definition.
    4. The paper does not clearly state the *mild conditions* (phrase used several times in the paper) under the Type-2 error is controlled.
    5. What is meant by the phrase "aligns with the correct answer"?
    6. What are $\mathcal{Q}$ and $\mathcal{A}$? They are used before definition in line 105.
    7. Line 269: How is the "probability distribution over distinct meanings" defined?
    8. What is FactTest-t that comes up Section 4.3, without any prior definition?
6. *Definition of M(q)*
    1. I sense ambiguity in the statement in line 116, where the authors mention that they consider the effects of the distribution of $M(q)$ as well in the probability term in equation 1. $M(q)$ is a certain realization from the distribution of responses, which is not explicitly captured in the expression. I would encourage the authors to explain this point more explicitly.
    2. All through sections 2 and 3, the framework used M(q) as a single generation for a given question q. However, in the experiments, in equation 6, M(q) appears to be a list of responses. I would encourage the authors to be consistent in their notations.
8. Major typos:
    1. I think that in line 167, it should be "ability that 𝑀 answers the question *𝑞* (not q') certainly given any question 𝑞′ and the generated answer 𝑀(𝑞′)".
    2. It looks like the legends of Figure 1 have typos in them, as the plot names are repeated.
    3. Line 465: Shouldn't it be FactTestO-SE instead of FactTest-SE?
11. Line 223: The authors should provide the proof for $\tilde{\mathcal{D}}_0\mid\mathcal{I}\sim P_0$ and for iid. Moreover, what is meant by the notation: $\tilde{\mathcal{D}}_0\mid\mathcal{I}$, specifically, the conditioning?
14. Experiments:
    1. The evaluation is just on Llama models and GPT-4o-mini. There are several other small open-source and closed-source models that must be evaluated to fully understand the efficacy of the method. Examples are Mistral, Gemini, Claude, GPT-4o, Phi, etc.
    2. I think the evaluations should also report the % of willingly answered questions. Without that, it is hard to judge whether the method makes the models too conservative about QA.
    3. Table 1 must also report the accuracy of the pretrained model on the subset of questions answered willingly in the FactTest experiments.
    4. For practical significance level $\alpha=0.05$, the Type-2 error shown in Figure 2 does not appear to be controlled. It appears to invalidate the claim of the paper about controlling Type-2 error too.
    5. The experiments consider only finetuning-based baselines and has no prior uncertainty quantification baselines or other hallucination mitigation methods to compare their uncertainty results against.
    6. It is not made clear what parts of the datasets used were for training and testing of the methods.
    7. The main text should mention how the ParaRel-OOD dataset differs from ParaRel.
    8. I don't understand how FactTest can work on just the pretrained model, without instruction tuning. The former models are known to not output the answer properly in most settings, which is the main motivation of instruction tuning. Why can't FactTest be applied to instruction tuned models?
25. The paper does not mention the relevant prior works on providing guarantees on the generations of LLMs. A non-exhaustive list is following:
    1. C-RAG: Certified Generation Risks for Retrieval-Augmented Language Models by Kang et al.
    2. Decoding Intelligence: A Framework for Certifying Knowledge Comprehension in LLMs by Chaudhary et al.
    3. Language Models with Conformal Factuality Guarantees by Mohri et al.

**Questions:**

1. What does the parameterization wrt $\mathcal{D}$ mean in the outermost probability term in Equation 1? The paper mentions $\mathcal{D}$ as a dataset, which could be the sample space of the distribution. So what distribution is this probability defined over?
4. Do $\mathcal{D}_0$ and $\mathcal{D}_1$ contain of multiple answers $M(q)$ for same $q$?
5. Is $(q',M(q'))$ in line 167 from the datasets, or *any* possible pair?
6. Lines 172-173: How does assuming H to be an identity function ensure the condition $\|H\circ\hat{\eta}-\eta\|_\infty\leq\epsilon_\eta$? What is the point of having H in the first place, then?
7. Line 201: can the target distribution not be redefined and hence expanded to account for the covariate shift? Hence the previous theory can be reused.
8. Lines 213-215: Do the source and target distributions have the same sample space?
9. Line 256: Why is the expected value of $\tilde{v}$ taken?
10. How is the frequency/probability term in Equation 6 calculated/estimated?
12. Lines 289-290: how will the distribution-free setting be violated for models requiring fine-tuning to answer factual question? I think most LLMs can do factual QA (perhaps not optimally) without finetuning. So what is the point of mentioning this?
13. Why does KLE have only a 15 generation variant in Table 1?
14. Does the experiment corresponding to Figure 3 suggest that after the construction of the certainty classifier (basically identification of its threshold), one needs to do another search for the accuracy maximizing threshold? I don't get the point of this experiment, if the search for the accuracy maximizing threshold is not a part of the method.
15. Lines 459-460: How do you train a classifier to approximate density ratios? Is it unsupervised training?
17. In the black-box APIs setting, is the open-source model used to get the uncertainty score even during testing?

---

> ### Author Response · Authors · 2024-11-21
> **Response to Reviewer Fj97 (1)**
>
> Thank for your constructive feedbacks. We are glad that you acknowledge the experiments of our work. Here we provide responses to your comments one by one. We also add more experimental results according to your suggestions. We hope these could address your concerns (also see the revised paper in the updated pdf):
>
> > ***W1: concerns about the equivalence of the notions of certainty of the model and its correctness.***
>
> **R1:** Thank you for your question. We want to clarify that our work does not rely on the premise that if the model is certain then it is going to be correct. FactTest can control the Type I error with any score function, while Type II error can be controlled if the score function indeed quantifies the model correctness. Given the difficulty of directly measuring correctness without ground truth labels during testing, we follow prior works and use uncertainty as an indicator of potential hallucination. In our implementation, we utilize the certainty scores to serve as score functions, which do not contradict previous works.
>
> In our previous PDF, we term 'the model answers q correctly' as 'model being certain of q', which may cause some confusion. Therefore, we have revised the expressions in Sec.1 and Sec.2 and differentiate these two terms. Some modifications are: (1) If null hypothesis is rejected, i.e., $M(q)$ aligns with $a$, the question-generated answer pair will be deemed correct; otherwise, it's incorrect. (2) $y_i$ indicates the correctness of $M(q)$, based on which the samples will be divided into incorrect subset $\mathcal{D}_0$ and correct subset $\mathcal{D}_1$.
>
> As for your concern about hallucination in Line 034, we want to rephrase the definition of it, which is *the generated content that nonsensical or unfaithful to the provided source content but **appears to be fluent and grounded in the real context.*** [1,2] The "seemingly high confidence" is different from the inherent model uncertainty to be estimated. We have modified the expression to *'generate nonfactual and incorrect information with seemingly high fluency and natural grounding'* in Line 034 to make it clearer.
>
> > ***W2: a mismatch in the definition of hallucination: the models generating incorrect responses with high confidence；hallucination occurs when model is uncertain.***
>
> **R2:** Thank you for your question. As stated in **R1**, the "seemingly high confidence" in our original version means the LLMs output hallucinations in a way that seems natural and fluent, which is hard to tell apart from other “real” perceptions [1]. This is different from the inherent model uncertainty. We follow the prior works and assume that when the model is uncertain, there's a high probability that the generated output is a hallucination. We have modify the statement regarding hallucination in Line 34 to make it more clearer.

---

> ### Author Response · Authors · 2024-11-21
> **Response to Reviewer Fj97 (2)**
>
> > ***W3: theoretical generalizability of the uncertainty predictor: useful as a general uncertainty calibrator？***
>
> **R3:** Thank you for the question. However, there seems to be a misunderstanding in this comment. To clarify, let us first explain the distribution we consider in this work. As we defined on Page 2, we assume there is a distribution $P_{q,a}$ over all the possible question-answer pairs $(q,a)$. The marginal distribution $P_q$ of $q$ is over all the possible questions, and the conditional distribution $P_{a|q}$ of $a$ given $q$ is supported on the set of (correct) answers to $q$. Therefore, under $P_{q,a}$, $a$ given $q$ can be viewed a random answer among all the correct answers of $q$. Recall that given any question $q$, the distribution $P_{M(q)|q}$ of the answer $M(q)$ generated by $q$ is fully determined by the language model $M$ and does not rely on the correct answer $a$ given $q$, which means $M(q)\perp a|q$. Following this, we defined a distribution $P_{q,M(q),a}=P_{q}P_{a|q}P_{M(q)|q}$ over all the possible combinations $(q,M(q),a)$. Then we introduce another binary random variable $y=\mathbb{I}(M(q)\text{ aligns with }a)$ indicating whether the generated answer $M(q)$ aligns with the correct answer $a$. Therefore, $y$ is deterministic given $q,M(q),a$. This construction results in a well defined distribution $P_{q,M(q),a,y}$. Finally as we defined on Page 3, we abbreviate the distribution $P_{q,M(q)|y=0}$ as $P_0$ and $P_{q,M(q)|y=1}$ as $P_1$.
>
> Equipped with the definition of $P_{q,M(q),a}$, we can view the dataset $\mathcal{D}=\lbrace(q_i,M(q_i),a_i):i\in[n]\rbrace$ as $n$ i.i.d. samples from $P_{q,M(q),a}$. After defining $y_i=\mathbb{I}(M(q_i)\text{ aligns with }a_i)$, the set $\mathcal{D_0}=\lbrace(q_i,M(q_i)):y_i=0,i\in[n]\rbrace$ given $\lbrace y_i:i\in[n]\rbrace$ can be viewed as $n_0=\sum_{i\in[n]}\mathbb{I}(y_i=0)$ i.i.d. samples from $P_0$, and similar for $\mathcal{D_1}$.
>
> Using the datasets introduced above, the type I and II error controls can be summarized as follows:
> 1) With probability at least $1-\delta$ over the randomness of the dataset $\mathcal{D}$, the probability that $\hat f_\alpha$ misclassifies any independent incorrect test sample $(q,M(q))$ from $P_0$ as correct is below $\alpha$.
> 2) With probability at least $1-2\delta$ over the randomness of the dataset $\mathcal{D}$, the probability that $\hat f_\alpha$ misclassifies any independent correct test sample $(q,M(q))$ from $P_1$ as incorrect is not too large, compared to that of the optimal classifier with controlled type I error.
>
> Since our goal is to detect incorrect answers for any future question-answering scenarios, not restricted to questions in the calibration data, the distribution we consider covers all possible question-answer pairs, including elements outside $\mathcal{D}$. Moreover, our method has error control over any independent question-answer pairs, therefore it can be used as a general calibrator for detecting incorrectness.

---

> ### Author Response · Authors · 2024-11-21
> **Response to Reviewer Fj97 (3)**
>
> > ***W4: a proof sketch for Theorem 1***
>
> **R4:** Thank you for the question. **The proof sketch is already provided in Appendix** due to the space limit. Here we provide more clarification to address your concern:
>
> The type II error control in Tong (2013)[3] is for a very different method. Their method takes the empirical type I error as a constraint. In order to control the population type I error at level $\alpha$, they constrain the empirical type I error at level $\alpha-c\sqrt{\frac{\log 1/\delta}{n_0}}$. This constaint also restricts the sample size $n_0$ to be large enough such that $\alpha\gtrsim\sqrt{\frac{\log 1/\delta}{n_0}}$, while **our type I error control works for any sample size $n$ and our type II error control only requires $\alpha\gtrsim\frac{\log (1/\delta)}{n_0}$**.
>
> Even **for the proof of type II error control, our method is also different** from that in Tong (2013). The excess type II error can be decomposed into two terms: the first term quantifies how conservative $\hat f_\alpha$ is in the type I error control, i.e., $\alpha-\mathcal{R_0}(\hat f_\alpha)$, and the second term corresponds to the estimation error of $\hat f_\alpha$ for the Bayes optimal classifier $f^*_\alpha$.
>
> For the second term, our analysis is not restricted to Holder class and only requires $\eta$ can be estimated up to increasing transformations, i.e., $\Vert H\circ\hat\eta-\eta\Vert_\infty$ (we will explain this useful condition further in the response to Question 4), while Tong (2013) assume $\eta$ is Holder smooth and can be estimated directly.
>
> Our analysis of the first term is also unique. The conservativeness in type I error control in Tong (2013) is due to the deviation between the empirical type I error and population type I error, which is straightforward to analysis. For our method, recall that in our construction of $\hat f_\alpha=\mathbb{I}(\hat\eta>\hat\tau_\alpha)$, the threshold $\hat\tau_\alpha$ is chosen from $n_0$ certainty scores $T_i=\hat\eta(q_i^{(0)},M(q_i^{(0)}))$, then the conservativeness of our method is due to the finite choices of thresholds. Our analysis for the first term relies on a detailed understanding of the behaviour of these thresholds.
>
> > ***W5: using terms before definition***
>
> **R5:** Thank you for pointing our the problem. We will clarify them in the main text.
>
> 1. Type I error, Type II error: The definition of Type I error **has already been defined in Line 013 and Line 045**. Since Type II error is opposite to Type I error, we omitted it in the original version, which has now been added to Sec 2.3. In our setting, Type I error is the probability of misclassifying incorrect $(q,M(q))$ from $P_0$ as correct. Type II error is the probability of misclassifying correct $(q,M(q))$ from $P_1$ as incorrect.
> 2. human-annotated samples: Human-annotated samples represent data samples with human-annotated labels, which in our setting is $\{(q_i,a_i):i\in[n]\}$ containing the set of $n$ questions $q_i$'s and corresponding correct answers $a_i$'s typically provided by humans.
> 3. $\epsilon_{\eta}$: $\epsilon_\eta$ is also defined (in line 170 in the original version), where we assume there exists an increasing function $H$ and $\epsilon_\eta\ge0$ such that $\Vert H\circ\hat\eta-\eta\Vert_\infty\le\epsilon_\eta$.
> 4. *mild conditions*  under which the Type II error is controlled: We have modified the expression in line 55 in the revised PDF to make it clearer. Specifically, we make the following three assumptions for type II error control: 1) $\alpha\gtrsim\frac{\log 1/\delta}{n_0}$ instead of $\sqrt{\frac{\log 1/\delta}{n_0}}$ required by Tong (2013)[3], 2) $\hat\eta(q,M(q))$ is a continuous random variable with $(q,M(q))\sim P_0$, 3) $\tau_\alpha+\epsilon_\tau+\epsilon_\eta<1$, where $\tau_\alpha$ is the threshold of the Bayes optimal classifer with type I error constrained below $\alpha$, and $\epsilon_\tau$ defined in line 184 is expected to be of small order.
> 5. aligns with the correct answer: Thank you for pointing out the issue. We have added details in Sec 4.1. Specifically, evaluating whether $M(q)$ aligns with the answer $a$ depends on the datasets. For question-answering datasets, we verify whether the first few output tokens contain $a$. For multiple-choice datasets, we check whether $M(q)$ exactly matches $a$.
> 7. $\mathcal{Q}$ and $\mathcal{A}$: $\mathcal{Q}$ is the set of all possible questions and $\mathcal{A}$ is the set of all possible answers. We have added it to the updated PDF.

---

> ### Author Response · Authors · 2024-11-21
> **Response to Reviewer Fj97 (3)**
>
> 7. probability distribution over distinct meanings: This is proposed by Semantic Entropy, and the details and equations can be seen in appendix. Specifically, $$
>     SE(q,M(q)) = - \sum_{c} p(c|q) \log p(c|q)
>     = -\sum_c \bigg(\Big(\sum_{\mathbf{a} \in c} p(\mathbf{a} \mid  q)\Big) \log \Big[ \sum_{\mathbf{a} \in c} p(\mathbf{a} \mid q) \Big]\bigg)
> $$
>     where $c$ represents possible meaning-class and $p(\mathbf{a}|q)$ is the probability of the entire answer sequence, that is, the product of the conditional probabilities of new tokens given past tokens.
> 8. FactTest-t: The definition of FactTest-t **is already defined in Section 4.1**. To facilitate comparison with training-based methods, we randomly split our training dataset, allocating half for instruction-tuning and the remaining half to construct the calibration dataset. We use 15-generation SE as the score function, referring to this variant as FactTest-t

---

> ### Author Response · Authors · 2024-11-21
> **Response to Reviewer Fj97 (4)**
>
> > ***W6: Definition of M(q)***
>
> **R6:** Thank you for your question. Given any question $q$, the answer $M(q)$ generated by language model $M$ is a random answer following distribution $P_{M(q)|q}$. The distribution $P_{M(q)|q}$ is fully determined by $q$ and $M$, but the random draw $M(q)|q$ involves the sampling randomness independent of $q$ and $M$. In the dataset $\mathcal{D}=\lbrace (q_i,M(q_i),a_i):i\in[n]\rbrace$, the observed $M(q_i)$ is a realization from $P_{M(q)|q=q_i}$. Then the logic behind our correctness predictor $\hat f_\alpha(q,M(q))=\mathbb{I}(\hat\eta(q,M(q))>\hat\tau_\alpha)$ is as follows.
>
> For any new question $\tilde q$, we ask $M$ to generate an output $M(\tilde q)$, then we aim to judge whether the current realization $M(\tilde q)$ is correct or not, based on the question $\tilde q$, the distribution $P_{M(q)|q=\tilde q}$, and the realization $M(\tilde q)$. If we think $M(\tilde q)$ is incorrect, we refuse question $\tilde q$.
>
> Note a special case of $\hat f_\alpha$ is that it ignores the realization $M(\tilde q)$ and make decision based on solely $\tilde q$ and $P_{M(q)|q=\tilde q}$. In other words, we judge whether the question $\tilde q_i$ is difficult for $M$. If we think $\tilde q$ is hard and $M$ is likely to generate incorrect answers, we refuse question $\tilde q$ regardless of the realization of the produced answer $M(q)$, although $M$ may still have some probability to produce correct answer.
>
> In Section 2 and 3, we use the more general form of $\hat f_\alpha$ to include $M(\tilde q)$ as an argument. But in the experiment, we consider the special case where $\hat f_\alpha$ make decision based on $\tilde q,P_{M(q)|q=\tilde q}$ and doesn't rely on the specific realization $M(\tilde q)$. The way we utilize $P_{M(q)|q=\tilde q}$ is through Monte-Carlo approximation by generating $k$ answers $\lbrace M(\tilde q)_j:j\in k\rbrace$ from
>
> $P_{M(q)|q=\tilde q}$.
>
> > ***W7: typos:***
>
> **R7:** Thank you for pointing out the potential typos:
> 1) The statement you modified is correct, but our statement is also correct. For generic random element $(q,M(q),a,y)\sim P_{q,M(q),a,y}$, we denote the $\eta(q,M(q))=ℙ(y=1|q,M(q))$, then for any fixed pair $(q',M(q'))$, $\eta(q',M(q'))=ℙ(y=1|q=q',M(q)=M(q'))$ can be interpreted as the conditional probability for $M(q)$ to align with $a$ given $(q,M(q))=(q',M(q'))$, which can of course be equivalently stated as the conditional probability for $M(q')$ to align with $a$ given $(q,M(q))=(q',M(q'))$. To avoid confusion, we change the statement in our paper and simply use $\eta(q,M(q))=ℙ_{y\sim P_{y|q,M(q)}}(y=1|q,M(q))$ instead of $\eta(q',M(q'))=ℙ_{(q,M(q),y)\sim P_{q,M(q),y}}(y=1|q=q',M(q)=M(q'))$.
> 2) Yes, thank you for pointing out. We have corrected it.
> 3) Yes, thank you for pointing out. We have corrected it.

---

> ### Author Response · Authors · 2024-11-21
> **Response to Reviewer Fj97 (5)**
>
> > ***W8: The proof for $\tilde{\mathcal{D}}_0 \mid \mathcal{I} \sim P_0$. What is meant by the notation: $\tilde{\mathcal{D}}_0 \mid \mathcal{I}$, specifically, the conditioning?***
>
> **R8:** Thank you for your suggestion. We have added the proof in our revision.
>
> Recall that $\mathcal{I}$ is an index set determined by the density ratios $w(q_i^{(0)},M(q_i^{(0)})$ and independent uniform random variables $U_i$. After determining the indices $\mathcal{I}$, for each index $i$, we can show that given $i$ is in $\mathcal{I}$, $(q_i^{(0)},M(q_i^{(0)}))$ follows $P_{q,M(q)|y=0}$. Consequently, given the index set $\mathcal{I}$, the samples with indices in $\mathcal{I}$ are i.i.d. samples from $P_{q,M(q)|y=0}$, i.e., $\tilde{\mathcal{D_0}}|\mathcal{I}\overset{i.i.d.}{\sim}P_{q,M(q)|y=0}$.
>
> Now we provide a proof for $(q,M(q))|\lbrace U\le w(q,M(q))\rbrace \sim P_{q,M(q)|y=0}=P_0$ with $(q,M(q))\sim\tilde P_0$. For any measurable set $C\subset\mathcal{Q}\times\mathcal{A}$, the conditional distribution of $(q,M(q))|\lbrace U\le w(q,M(q))\rbrace$ can be expressed as
> \begin{align}
> & \mathbb{P} ((q,M(q))\in C|U\le w(q,M(q))) \\\\
> = & \frac{\mathbb{P}((q,M(q))\in C, U\le w(q,M(q)))}{\mathbb{P}(U\le w(q,M(q)))} \\\\
> = & \frac{\mathbb{E}\frac{w(q,M(q))}{B}\mathbb{I}((q,M(q))\in C)}{\mathbb{E}\frac{w(q,M(q))}{B}} \\\\
> = & \mathbb{P}\_{(q,M(q))\sim P_{q,M(q)|y=0}}((q,M(q))\in C),
> \end{align}
> where we have use the facts that $\mathbb{P}(U\le w(q,M(q))|q,M(q))=\frac{w(q,M(q))}{B}$, $\mathbb{E}\_{(q,M(q))\sim \tilde P_0}w(q,M(q))=1$ and $\mathbb{E}\_{(q,M(q))\sim\tilde P_0}w(q,M(q))\mathbb{I}((q,M(q))\in C)=\mathbb{P}_{(q,M(q))\sim P_0}((q,M(q))\in C)$.
>
> > ***W9-1: (Experiments)The evaluation is just on Llama models and GPT-4o-mini.***
>
> **R9-1:** Thank you for your advice. Due to the time and resource limits, we only included Llama models and GPT-4o-mini before. We have now added the experiments on Mistral and other closed-source models including GPT-4o, Gemini and Claude. The results are shown as follows and we have included these new results into our paper(See Sec.E.4 in our revised PDF). Hope that these new experiment results can help better understand the efficacy of the method.
>
> Table 1: The accuracy performance of FactTest on four question-answering datasets using Mistral-7B as the base model. The significance level for FactTest is set to 0.1. The percentages inside the parentheses are the Type I error.
>
> | Dataset  | Pretrained | SelfCheckGPT-NLI    | FactTest-ve15 | FactTest-se15 | FactTest-kle15 |
> | -------- | ---------- | --- | ------------- | ------------- | -------------- |
> | ParaRel  | 39.79      | 57.01 (0.25)  | 65.63 (0.07)   | 70.20 (0.08)   | 72.78 (0.08)    |
> | HotpotQA | 36.48      | 46.01 (0.46) | 61.81 (0.06)   |   63.06 (0.05)            | 65.59 (0.05)    |
> | FEVER    | 35.47      |  41.76 (0.05)   |   22.99 (0.08)            | 51.05 (0.08)   | -              |
> | WiCE     | 55.85      |   56.24 (0.47)  | 68.81 (0.08)   | 68.64 (0.08)   | -              |
>
> Table: The accuracy performance of FactTest on ParaRel using llama 7B as open-source model. The significance level is set to 0.1. The percentages inside the parentheses are the Type I error.
> | Model  | Base  | SelfCheckGPT-NLI | FactTest-se15 | FactTest-kle15 |
> | ------ | ----- | ---------------- | ------------- | -------------- |
> | Claude-3.5-Sonnet | 58.25 | 58.96 (0.92)     | 73.29 (0.08)  | 79.86 (0.08)   |
> | Gemini-1.5-Flash-8B   | 64.23  |  65.92 (0.86)  |         76.87 (0.07)     |  80.01 (0.08)    |
> | GPT-4o | 66.39 | 69.71 (0.83)     | 80.70 (0.07)  | 82.76 (0.08)   |

---

> ### Author Response · Authors · 2024-11-21
> **Response to Reviewer Fj97 (6)**
>
> > ***W9-2: (Experiments) The evaluations should also report the % of willingly answered questions.***
>
> **R9-2:** Thank you for your constructive advice. The percentage of the willingly answered questions will vary with different significance level $\alpha$, the allowable probability $\delta$, the score functions, base models and datasets. One could modify significance levels for a balance between being conservative or aggressive in answering questions.
>
> Here we provide an answer rate analysis of our method using different significance levels comparing with baselines, which has been added to our modified paper in Sec E.3.
>
> Table: The Answer Rate and Accuracy Performance (%) of FactTest-t. The number in parenthese is the percentage of willingly answered questions.
>
>
> | Dataset | Model | Finetuned  | R-Tuning |FactTest-t ($\alpha$=0.15) |  FactTest-t ($\alpha$=0.1)  | FactTest-t ($\alpha$=0.05) |
> | -------- | -------- | --- | --- | --- | --- | -------- |
> | ParaRel  | OpenLLaMA-3B |61.73 ( 100% ) | 87.42 ( 37% ) | 89.91 ( 46% )  |92.73 ( 31% )| 94.26 ( 17% )  |
> |          |  LLaMA-7B   | 67.73( 100% ) | 89.65 ( 42% )    |   92.76 ( 47% )  | 95.04 ( 31% ) |  96.01 ( 18% )  |
> | FEVER  | OpenLLaMA-3B  | 65.56 ( 100% ) |  67.19 ( 11% ) |   92.58 ( 38% )   | 94.88 ( 36% )  |  97.82 ( 33% )   |
> |        | LLaMA-7B  | 66.24 ( 100% ) |  66.19 ( 49% )  | 95.41 ( 28% )    |  95.83 ( 24% )    | 96.79 ( 16% ) |
>
> The findings demonstrate that **FactTest consistently achieves higher accuracy while effectively managing the answer rate through varying significance levels**. Specifically, FactTest-t with $\alpha=0.15$ answers 47% questions on ParaRel and acheives 92.76% accuracy, outperforming R-Tuning, which answers 42\% of the questions with an accuracy of 89.65\%. Similarly, FactTest-t maintains superior accuracy performance on FEVER compared to baseline models while managing the answer rate through different significance levels.
>
> > ***W9-3: (Experiments) Table 1 must also report the accuracy of the pretrained model on the subset of questions answered willingly in the FactTest experiments.***
>
> **R9-3:** Thank you for your question, but there may be a misunderstanding. The accuracies of the pretrained models on the subset of willingly answered questions **are the results of FactTest**. It can be applied to all kinds of LMs including pretrained models and instruction-tuned models(e.g. Tulu, Llama-Instruct) and identifies the questions that the LM cannot provide correct answers.
> Here we additionally present the accuracy of pretrained models on the subset of questions that the model is unwilling to answer on ParaRel using FactTest-kle15 to supplement the results in main text, which has been added to our updated PDF in Sec E.6. The $\alpha$ is set to 0.1.
>
> | Model         | Pretrained |  Unwilling   | Willing |
> | ------------- | ---------- | --- | --------- |
> | Openllama-3B  | 36.66      |  27.90   | 75.51   |
> | Openllama-7B  | 40.38      |  32.93   | 75.36   |
> | Openllama-13B | 42.21      |   32.81  | 79.55   |
>
> > ***W9-4: (Experiments) For $\alpha$=0.05, the Type-2 error shown in Figure 2 does not appear to be controlled.***
>
> **R9-4:** Thank you for your question. We acknowledge that in some instances, the Type II error may not appear to be adequately controlled. However, this will not violate what we prove in our Type II error control analysis, which is based on the premise that the score function effectively measures the correctness of the generated answers. We have theoretically established that the optimal classifier for minimizing Type II error, given a constraint on Type I error, adopts a thresholding rule based on an oracle score. In practice, since the oracle score is inaccessible, we rely on a certainty function to approximate it. Our theoretical guarantees assert that if this score function approximates the oracle score well, up to an increasing transformation, then the Type II error will be effectively controlled. More critically, if the score function fails to accurately assess the correctness of the generated answers, though our Type I error control holds for any score functions, the Type II error control may falter.

---

> ### Author Response · Authors · 2024-11-21
> **Response to Reviewer Fj97 (7)**
>
> > ***W9-5: (Experiments) no prior uncertainty quantification baselines or other hallucination mitigation methods.***
>
> **R9-5:** Thank you for your suggestion. In fact, all uncertainty quantification methods can serve as the score functions and be integrated into our framework. Besides, hallucination mitigation tries to improve the model outputs while our goal is to detect hallucination. We now include SelfCheckGPT-NLI[4] as our baseline, which is a zero-resource hallucination detection method. It will output a contradiction probability between 0 and 1, and then we evaluate the answers to questions with a score less than 0.5. We list some results as follows, and for the complete table please refer to Table 1 in the updated PDF.
>
>
> Table: The accuracy performance (%) of FactTest with significance level = 0.5.
> | Dataset  | Model         | SelfCheckGPT-NLI | FactTest-ve15 | FactTest-se15 | FactTest-kle15 |
> | -------- | ------------- | ---------------- | ------------- | ------------- | -------------- |
> | ParaRel  | OpenLLaMA-3B  | 53.60            | 67.28         | 67.26         | 78.45          |
> |          | OpenLLaMA-7B  | 60.05            | 80.29         | 65.23         | 76.83          |
> |          | OpenLLaMA-13B | 59.62            | 79.41         | 73.09         | 83.84          |
> | HotpotQA | OpenLLaMA-3B  | 36.42            | 53.75         | 52.66         | 55.35          |
> |          | OpenLLaMA-7B  | 39.16            | 60.67         | 56.56         | 60.66          |
> |          | LLaMA-13B     | 41.78            | 49.74         | 60.69         | 54.49          |
> | WiCE     | OpenLLaMA-3B  | 66.36            | 68.18         | 66.67         | -              |
> |          | OpenLLaMA-7B  | 75.00            | 47.37         | 90.00         | -              |
> |          | LLaMA-13B     | 57.39            | 44.44         | 90.00         | -              |
> | FEVER    | OpenLLaMA-3B  | 41.97            | 41.72         | 83.90         | -              |
> |          | LLaMA-7B      | 40.89            | 51.38         | 33.27         | -              |
> |          | LLaMA-13B     | 41.25            | 46.07         | 52.23         | -              |
>
> In fact, SelfCheckGPT can also be integrated into our framework. To do this, we can use the neative value of SelfCheckGPT-NLI contradiction score as a score function, termed as FactTest-scgpt. We provide accuracy and Type I error results on ParaRel with llama 3b.
>
>
> | $\alpha$     | 0.05  |  0.1   | 0.2   |
> | ------------ | ----- | --- | ----- |
> | Accuracy     | 61.82 |  62.92   | 59.26 |
> | Type 1 error | 0.04  |  0.09   |  0.17  |
>
>
> > ***W9-6: It is not made clear what parts of the datasets used were for training and testing of the methods.***
>
> **R9-6:** We follow the same train/test split of the datasets in [5] except for FactTest-t. As for FactTest-t, the training dataset is then be divided randomly into two equal parts for finetuning and calibration, which is introduced in Sec.4.3. We have added details in our appendix.
>
> > ***W9-7: The main text should mention how the ParaRel-OOD dataset differs from ParaRel.***
>
> **R9-7:** Thank you for your suggestions, we have added a brief introduction in Sec 4.4 (Line 264).
>
> > ***W9-8: Why can't FactTest be applied to instruction tuned models?***
>
> **R9-8:** Thank you for your question and advice. Actually, FactTest can be applied to any LMs including pretrained models and instruction tuned models. The reason why we didn't provide experiments on instruction tuned models is due to time and space limits. To make the experimental results more convincing, we now evaluate FactTest on two popular instruction-tuned models Llama3.2-3B-Instruct and Tulu-2-7B. We also add these results to the appendix in the revised PDF. Results show that FactTest works well on instruction-tuned models, improving the accuracy performance while effectively controling the Type 1 error.
>
> Table: The accuracy performance and Type 1 error of FactTest using instruction-tuned models as base models. The significance level is set to 0.1.
>
> | Dataset  | Model                 | Base  | FactTest-se15 | FactTest-kle15 |
> | -------- | --------------------- | ----- | ------------- | -------------- |
> | ParaRel  | Llama-3.2-3B-Instruct | 39.34 | 72.79 (0.08)   | 80.01 (0.08)    |
> | ParaRel  | Tulu-2-7B             | 43.89 | 75.47 (0.06)   | 78.49 (0.07)    |
> | HotpotQA | Llama-3.2-3B-Instruct | 33.40 | 57.75 (0.06)          | 60.38 (0.07)   |
> | HotpotQA | Tulu-2-7B             | 32.91 | 53.54(0.05)   | 45.89(0.10)    |
> | WiCE     | Llama-3.2-3B-Instruct | 55.11 | 75.16 (0.09)   | -              |
> | WiCE     | Tulu-2-7B          |  57.20  |  63.22 (0.08)             | -              |
> | FEVER    | Llama-3.2-3B-Instruct | 33.33 | 68.48 (0.10)   | -              |
> | FEVER    | Tulu-2-7B             | 47.87 | 69.40 (0.09)   | -              |

---

> ### Author Response · Authors · 2024-11-21
> **Response to Reviewer Fj97 (8)**
>
> > ***W10: mention the relevant prior works on providing guarantees on the generations of LLMs.***
>
> **R10:** Thank you for your suggestions. We have included prior works on providing guarantees on the generations of LLMs as well as the differences between FactTest and these works in our updated PDF(See Sec.5 and Sec.B). Here we provide a brief summary about the related works you mentioned:
>
> C-RAG [6] provides conformal risk analysis for RAG models and certifies an upper confidence bound. Conformal Factuality [7] enables the application of conformal prediction in improving model performance while FactTest evaluates the correctness and abstain from answering unknown questions. QuaCer-C [8,9] certifies knowledge comprehension in LLMs with formal probabilistic guarantees, whose goal is similar to ours but it only focuses on knowledge comprehension task.

---

> ### Author Response · Authors · 2024-11-21
> **Response to Reviewer Fj97 (9)**
>
> > ***Q1: What does the parameterization wrt $\mathcal{D}$ mean in the outermost probability term in Equation 1?  What distribution is this probability defined over?***
>
> **R1:** As we have explained in the response to Weakness 3 (W3), samples in $\mathcal{D}$ follow the distribution $P_{q,M(q),a}$, then the two probabilities in Equation (1) can be understood in the following way.
> 1) The inner probability $\mathbb{P}\_{(q,M(q))\sim P_0}$ counts the randomness of the independent incorrect test sample $(q,M(q))\sim P_0$ and the classifier $\hat f_\alpha$ is fixed here.
> 2) The outer probability $\mathbb{P}\_{\mathcal{D}}$ is taken with respect to the randomness of the dataset $\mathcal{D}\overset{i.i.d.}{\sim}P_{q,M(q),a}$, or equivalently, $\mathbb{P}\_{\mathcal{D}}$ counts all the randomness in the classifier $\hat f_\alpha$.
>
> > ***Q2: Do $\mathcal{D_0}$ and $\mathcal{D_1}$ contain of multiple answers $M(q)$ for same $q$?***
>
>
> **R2:**
> No. Recall $\mathcal{D_0}=\lbrace (q_i,M(q_i)):y_i=0,i\in[n]\rbrace$ and $\mathcal{D_1}=\lbrace (q_i,M(q_i)):y_i=1,i\in[n]\rbrace$ contain all the incorrect samples and correct samples, respectively. In $\mathcal{D}\_0$ and $\mathcal{D}\_1$, $M(q_i)$ is the currect realization of the answer for $q_i$ produced by $M$.
>
> If we ask a language model $M$ a question $q$ once, it only output one answer $M(q)$, and our goal is to judge whether the currect output $M(q)$ is correct or not. To this end, we collect $n$ realizations of this question-answering procedure and aim to learn some common rules from the data.
>
> > ***Q3: Is $(q',M(q'))$ in line 167 from the datasets, or any possible pair?***
>
> **R3:** $(q',M(q'))$ can be any possible question-generated answer pair, not restricted to the observed samples.

---

> ### Author Response · Authors · 2024-11-21
> **Response to Reviewer Fj97 (10)**
>
> > ***Q4: How does assuming H to be an identity function ensure the condition $\Vert H\circ\hat{\eta}-\eta\Vert_\infty\leq\epsilon_\eta$? What is the point of having H in the first place, then?***
>
> **R4:**
> In our paper, we used the sentence "WLOG, we assume $H$ is the identity function" to simplify the notations and derivations in the type II error analysis. This simplification mainly works for the proof of Theorem 2 and does not affect the statement of this theorem. In the revision, we move this statement into the appendix to avoid confusion.
>
> In the following, we will explain the role of the increasing transformation $H$ and the validity of this "WLOG" simplification. Specifically, we will demonstrate in the following paragraphs that 1) our construction of $\hat f_\alpha$ is invariant under increasing transformations of $\hat\eta$, then we can pretend that the score function is $H\circ\hat\eta$ and thus, the transformation becomes identity for the new score, 2) the introduction of $H$ allows a more flexible metric to quantify the difference between $\hat\eta$ and $\eta$, allowing the usage of many modern classification algorithms for training a score function from data.
>
> Suppose there exists some increasing function $H$ such that $\Vert H\circ\hat\eta-\eta\Vert_\infty\le\epsilon_\eta$. Recall that the classifier we consider has the form $\hat f_\alpha=\mathbb{I}(\hat\eta>T_{(\hat k)})$, where the index $\hat k$ satisfies Equation (5). If we replace $\hat\eta$ by $H\circ\hat\eta$ and rerun the algorithm using the new score function $H\circ\hat\eta$, we have the following observations:
> 1) $\hat k$ defined in Equation (5) remains the same.
> 2) The new classifier is $\mathbb{I}(H\circ\hat\eta>(H(T))\_{(\hat k)})$, where $H(T_i)=H\circ\hat\eta(q_i\^{(0)},M(q_i\^{(0)}))$ are the new scores of the samples and $(H(T))\_{(\hat k)}$ is the $\hat k$-th order statistic of $\lbrace H(T_i):i\in[n_0]\rbrace$ with $(H(T))\_{(1)}\le\ldots\le(H(T))\_{(n_0)}$.
> 3) Since $H$ is an increasing function, we have $(H(T))\_{(\hat k)}=H(T\_{(\hat k)})$.
> 4) Then the new classifier $\mathbb{I}(H\circ\hat\eta>(H(T))\_{(\hat k)})$ equals $\mathbb{I}(H\circ\hat\eta>H(T\_{(\hat k)}))$ which further reduces to the original classifier $\hat f_\alpha=\mathbb{I}(\hat\eta>T\_{(\hat k)})$.
>
> These observations tell us that our algorithm is invariant under increasing transformation of the score function $\hat\eta$ and using $\hat\eta$ and using $H\circ\hat\eta$ lead to the same decision. Therefore, without the loss of generality, we pretend that we are using the score function $H\circ\hat\eta$ and increasing transformations are no longer required for this score function. This is the reason we assume $H$ is identity function.
>
> However, introducing the increasing transformation $H$ is extremely useful and is also a unique contribution of our work. Because if we train the score function using data, it corresponds to the probability of predicting $y=1$ based on $(q,M(q))$. It is well known that modern classification algorithms like deep neural networks are bad in calibration, which means they can not estimate the underline conditional probability $\eta(q,M(q))$ well. But our algorithm is still guaranteed to have small type II error if the deep neural networks are order consistent, in the sense that the two events $\lbrace\hat\eta(q_1,M(q_1))>\hat\eta(q_2,M(q_2))\rbrace$ and $\lbrace\eta(q_1,M(q_1))>\eta(q_2,M(q_2))\rbrace$ are close to each other. This flexibility allows the use of many modern classifiers for learning $\eta$.
>
>
> > ***Q5: Line 201: can the target distribution not be redefined and hence expanded to account for the covariate shift? Hence the previous theory can be reused.***
>
> **R5:** Yes, the previous theory can be reused after one additional step.
>
> The previous theory requires the data comes from the target distribution. In order to deal with covariate shift, we adopt rejection sampling, which aims to transform the data we have into samples from the target distribution. Once we get the samples from the target distribution, previous theory can be applied to guarantee the performance of our algorithm.
>
> > ***Q6: Lines 213-215: Do the source and target distributions have the same sample space?***
>
> **R6:** Yes, we do assume the support of target distribution is contained in that of the source distribution.

---

> ### Author Response · Authors · 2024-11-21
> **Response to Reviewer Fj97 (11)**
>
> > ***Q7: Line 256: Why is the expected value of $\tilde{v}$ taken?***
>
> **R7:** Recall that $\tilde v(k)=\sum_{j=k}^{\tilde n_0}{\tilde n_0\choose j}(1-\alpha)^j\alpha^{\tilde n_0-j}$, where $\tilde n_0$ is the size of incorrect samples selected by rejection sampling. As we have explained in the response to Question 5, the theoretical guarantee under covariate shift consists of two steps. Firstly, we apply rejection sampling to transform the calibration data into samples $\tilde{\mathcal{D}}\_0$ from the target distribution. Then, we reuse the previous theory to $\tilde{\mathcal{D}}\_0$ to conclude the result. Following this line, the equation you referred to can be interpreted as follows. Recall $\mathcal{I}$ is the index set selected using rejection sampling, as we have explained in the response to Weakness 8, $\tilde{\mathcal{D}}\_0|\mathcal{I}\overset{i.i.d.}{\sim}P_0$, then
> \begin{align}
> &\mathbb{P}\_{\mathcal{D}}\big(\mathbb{P}\_{(q,M(q))\sim P_0}(\hat\eta(q,M(q))>\tilde T_{(\hat k)})>\alpha\big)\\\\
> =&\mathbb{E}\_{\mathcal{I}}\mathbb{P}\_{\tilde{\mathcal{D}}\_0}\big(\mathbb{P}\_{(q,M(q))\sim P_0}(\hat\eta(q,M(q))>\tilde T\_{(\hat k)})>\alpha|\mathcal{I}\big)\\\\
> \le&\mathbb{E}\_{\mathcal{I}}\tilde v(\hat k)\\\\
> \le&\delta,
> \end{align}
> where on the right-hand side of the first equation, the inner probability $\mathbb{P}\_{\tilde{\mathcal{D}}_0}(\cdot|\mathcal{I})$ treats the samples $\tilde{\mathcal{D}}_0|\mathcal{I}$ selected by rejection sampling as samples from $P_0$, and reuse the previous type I error result in following inequalities, then, the outer expectation $\mathbb{E}\_{\mathcal{I}}$ counts the randomness due to rejection sampling.
>
> > ***Q8: How is the frequency/probability term in Equation 6 calculated/estimated?***
>
> **R8:** The frequency of a predicted answer $M(q)_j$ in Equation 6 is calculated by $\frac{m}{k}$, where $m$ is the number of times $M(q)_j$ exists in $k$ generations.
>
> We have added details about the calculation of frequency in Section B.2 updated PDF.
>
> > ***Q9: Lines 289-290: how will the distribution-free setting be violated for models requiring fine-tuning to answer factual question? I think most LLMs can do factual QA (perhaps not optimally) without finetuning. So what is the point of mentioning this?***
>
> **R9:** 'Distribution-free' refers to models or methods that do not make specific assumptions about the underlying probability distribution. In our main experiments, we utilize calibration dataset to provide distribution-free guarantees. Models trained on this dataset will be adjusted based on the data, and thus is not distribution-free. Therefore, it'll be unfair in main experiments to compare our method with finetuning-based methods.
>
> > ***Q10: Why does KLE have only a 15 generation variant in Table 1?***
>
> **R10:** Due to time and space limits, we only include a 15-generation variant in our main table. Theoretically，it should has a better performance than 10-generaion and 5-generation variants because the uncertainty estimation should be more accurate with more generations. We provide experiment results for 5-generation and 10-generation variants as follows, which has been updated in Sec D.5 in the updated PDF:
>
>
> Table: The Accuracy Performance and Answer Rate of FactTest with a significance level $\alpha=0.1$
> | Dataset | Model        | Base | FactTest-kle5 | FactTest-kle10 |
> | ------- | ------------ | ---------- | ------------- | -------------- |
> | ParaRel | OpenLLaMA-3B | 36.66      | 71.65 (18%)   | 74.72 (20%)    |
> |         | OpenLLaMA-7B | 40.38      | 72.99 (20%)   | 75.90 (20%)    |
> | Hotpot  | OpenLLaMA-3B | 25.72      | 52.34 (11%)   | 51.82 (12%)    |
> |         | OpenLLaMA-7B | 28.63      | 52.45 (11%)   | 55.92 (13%)    |
>
> Table: The Type I Error of FactTest with a significance level $\alpha=0.1$
> | Dataset | Model        | FactTest-kle5 | FactTest-kle10 |
> | ------- | ------------ | -------- | -------- |
> | ParaRel | OpenLLaMA-3B | 0.0783 | 0.0778  |
> |         | OpenLLaMA-7B | 0.0880  |  0.0787  |
> | Hotpot  | OpenLLaMA-3B |    0.0656      |   0.0643       |
> |         | OpenLLaMA-7B |   0.0643   |  0.0654  |
>
> > ***Q11: the experiment corresponding to Figure 3***
>
> **R11:** Sorry for the confusion. This experiment shows the variation tendency of accuracy with the threshold. With a user-specified $\alpha$, our framework can control the Type I error, which changes monotonically. However, the accuracy doesn't follow this monotonous trend, and the maximum accuracy depends on the model and score functions. The results provide insights for users to choose the sigficance level $\alpha$ as well as the performance of different score functions.

---

> ### Author Response · Authors · 2024-11-21
> **Response to Reviewer Fj97 (12)**
>
> > ***Q12: Lines 459-460: How do you train a classifier to approximate density ratios? Is it unsupervised training?***
>
> **R12:** We randomly split 1000 samples from ParaRel-OOD as validation samples and the remaining 12k samples as testing samples. We then utilize the supervised identification strategy to divide the validation samples into $D_0^{'}$ and $D_1^{'}$, and the training dataset into $D_0$ and $D_1$.
>
> We extract the features from the questions in $D_0^{'}$, $D_0$, and label them as 1(target data) and 0(source data). We then train a binary classifer and utilize the predicted probability to approximate density ratios.
>
> We have added the details in Sec C.3 in the updated PDF.
>
> > ***Q13: In the black-box APIs setting, is the open-source model used to get the uncertainty score even during testing?***
>
> **R13:** Yes. In Table 3, we utilize open-source model to provide certainty scores both in calibrating and testing. However, one could employ black-box uncertainty quantification methods to serve as score functions, which will not necessitate open-source models.
>
> We here provide another experiment using only black-box APIs to calculate scores. Specifically, we utilize the SelfCheckGPT with NLI score to serve as the score function, with significance level $\alpha$ = 0.1:
>
> |        | Base (Acc%) | FactTest-scgpt (Acc%) |
> | ------ | ----------- | --------------------- |
> | Claude |    58.25    |      75.54            |
> | GPT-4o |    66.39    |      73.65            |
>
> However, black-box uncertainty quantification methods usually prompt the APIs multiple times to compute uncertainty, which leads to much higher cost. Therefore, utilizing open-source models to calculate scores is a feasible and cost-friendly option.
>
> **Reference**
> [1] Survey of Hallucination in Natural Language Generation, ACM Computing Surveys 2023.
>
> [2] A Survey on Hallucination in Large Language Models: Principles, Taxonomy, Challenges, and Open Questions, 2023.
>
> [3] A plug-in approach to Neyman-Pearson classification, JMLR 2013.
>
> [4] SelfCheckGPT: Zero-Resource Black-Box Hallucination Detection for Generative Large Language Models, EMNLP 2023.
>
> [5] R-Tuning: Instructing Large Language Models to Say ‘I Don’t Know’, NAACL 2024.
>
> [6] C-RAG: Certified Generation Risks for Retrieval-Augmented Language Models, ICML 2024.
>
> [7] Language Models with Conformal Factuality Guarantees, ICML 2024.
>
> [8] Decoding Intelligence: A Framework for Certifying Knowledge Comprehension in LLMs, ArXiv 2024.
>
> [9] Quantitative Certification of Knowledge Comprehension in LLMs, SeT LLM @ ICLR 2024.

---

> ### Author Response · Authors · 2024-11-24
> **We would like to hear back from reviewer Fj97**
>
> Dear reviewer Fj97,
>
> We would like to follow up to see if the response addresses your concerns or if you have any further questions. We would really appreciate the opportunity to discuss this further if our response has not already addressed your concerns. Thank you again!

---

> > ### Comment · Reviewer_Fj97 · 2024-11-25
> >
> > Thanks to the authors for their rebuttal and additional experiments which clarify several of my previously mentioned concerns. I have, however, some concerns which are not convincingly addressed, mentioned below.
> > 1. I am confused by this statement "our work does not rely on the premise that if the model is certain then it is going to be correct". The entire paper relies on this premise and makes this assumption, including in the theory and experiments. I would recommend the authors scope their work to be about better uncertainty quantification than relate their uncertainty measure further to factuality.
> > 2. While the method assumes iid samples from some input distributions, experiments assume prior datasets to be iid and directly use them for analysis. It would have been ok for another paper that wouldn't claim guarantees. But for a work providing guarantees, such assumptions need to properly substantiated.

---

> > > ### Author Response · Authors · 2024-11-27
> > > **We would like to hear back from reviewer Fj97**
> > >
> > > Dear reviewer Fj97,
> > >
> > > Given the approaching revision deadline, we would welcome your assessment of whether our response adequately addresses your concerns. Please let us know if you need any clarification or have additional questions. Thank you again!

---

> ### Author Response · Authors · 2024-11-25
> **Response to Reviewer Fj97**
>
> Thank you for your follow-up comment. We appreciate the opportunity to clarify our methodology and address your concerns in greater detail:
>
> > The entire paper relies on this premise and makes this assumption, including in the theory and experiments.
>
> We would like to clarify that we said "our work does not rely on the premise that if the model is certain then it is going to be correct" in order to highlight that our theoretical framework ensures Type I error control regardless of the choice of score function. The score function could represent uncertainty, correctness, or even remain constant across all inputs. For instance, in the extreme case where the score function is a constant, Type I error can still be controlled by rejecting all questions, thereby maintaining valid statistical control of Type I error.
>
> However, controlling the Type II error does depend on the score function's ability to effectively quantify correctness. In our experiments, we primarily employed uncertainty-based measures as score functions because directly assessing correctness is inherently challenging. Nevertheless, our framework is not limited to uncertainty-based approaches. To illustrate this flexibility, we trained a random forest classifier to predict the correctness of question-answer pairs, using the predicted probability of the "correct" class as the score function. We refer to this approach as FactTest-cls and have included it in our updated manuscript. We compare FactTest-cls with two uncertainty-based variants, FactTest-ve15 and FactTest-se15, which employ entropy across generated answers and entropy incorporating linguistic invariances, respectively, to quantify uncertainty.
>
> The results are shown as follows (also see Table 13 in Sec D.5 for further details), which indicate that FactTest-cls achieves competitive accuracy and maintains Type I error below the specified threshold, while also demonstrating improved Type II error rates compared to uncertainty-based score functions.
>
> Table: The Accuracy, Type I error and Type II error performance of FactTest-cls compared with uncertainty-based score functions on ParaRel with $\alpha=0.05$.
>
> | Base Model   | Metric        | FactTest-ve15 |  FactTest-se15   | FactTest-cls  |
> | ------------ | ------------- | ------------- | --- | ------------- |
> | OpenLlama-3B | Accuracy(\%)  |   67.28    | 67.26  |         **85.13**      |
> |              | Type I error  |      0.05         | 0.05    |        0.04       |
> |              | Type II error |       0.86        | 0.85    |         **0.35**     |
> | OpenLlama-7B     | Accuracy(\%)  |   80.29   | 65.23 |        **89.50**       |
> |              | Type I error  |    0.01           |  0.04   |       0.03        |
> |              | Type II error |        0.92       |  0.87   |     **0.44**   |
> | OpenLlama-13B    | Accuracy(\%)  |  79.41  |  73.09   |       **88.37**        |
> |              | Type I error  |      0.03         | 0.03    |        0.04       |
> |              | Type II error |     0.91          |    0.87 | **0.42**       |
>
>
> > i.i.d. assumptions need to properly substantiated.
>
> Thank you for your comment. i.i.d. assumptions are commonly assumed in machine learning theory literature, ranging from generalization error bounds to conformal prediction (where they assume a related but slightly weaker assumption, exchangeablity) [2,3,4].
>
> In our experiments, we adhere to this assumption as follows:
>
> - 1). For ParaRel, we follow the setup in [1], dividing the dataset into two subsets: an in-domain subset, consisting of samples from the first 15 domains, and an out-of-domain (OOD) subset, comprising samples from the remaining 16 domains. The in-domain subset is then randomly split into training and testing sets, which ensures the data are i.i.d. The OOD subset is referred to as ParaRel-OOD and is utilized for evaluation with covariate shifts, which do not need to be i.i.d.
>
> - 2). For other datasets, we utilize standard training and testing splits, which are explicitly designed to follow the same underlying distribution. This setup adheres to the i.i.d. assumption required for our theoretical guarantees.
>
> We acknowledge that the i.i.d. assumption may not hold in certain cases. To address this, we have included a section in the paper extending our theoretical framework to the covariate shift setting in Sec.3, where the assumption is relaxed. We also plan to explore extensions to other types of distribution shifts in future work.
>
> References:
>
> [1] R-Tuning: Instructing Large Language Models to Say ‘I Don’t Know’, NAACL 2024.
>
> [2] Foundations of Machine Learning, The MIT Press 2018.
>
> [3] Conformal Prediction: A Gentle Introduction, Foundations and Trends in Machine Learning 2023.
>
> [4] Conformal Language Modeling, ICLR 2024.

---

> > ### Comment · Reviewer_Fj97 · 2024-12-01
> >
> > Thanks to the authors for their response. I believe that FactTest-cls is a good addition to the paper and helpful to mitigate some of my concerns. I still do not understand why the authors say that "assessing correctness is inherently challenging" when all QA benchmarks are about factuality and some of the related frameworks for statistical guarantees for LLMs (e.g., QuaCer-C from the paper) also evaluate response correctness, rather than uncertainty.
> >
> > About the iid assumption, the QuaCer-C paper from the related works section seems to be tackling a similar problem without any iid assumptions. Hence, I am not convinced about the utility of the guarantees of this work over those of QuaCer-C, which in my understanding can be extended to this particular setting of factuality.
> >
> > Overall, I think that this paper needs more work. The results shown in the rebuttal and further discussions are promising, but the claims need to be made more formal and informative, with proper specification of their scope. The original submission had numerous major statements and claims that I highlighted in my review that the authors have reverted now. Hence, I believe this paper needs another revision, consisting of proper positioning of the paper and its methods, before acceptance.

---

> > > ### Author Response · Authors · 2024-12-03
> > >
> > > Thank you for your response. We appreciate the opportunity to address your concerns and clarify aspects of our methods.
> > >
> > > We want to clarify more about "assessing correctness is inherently challenging": Unlike training or validation phases where ground-truth labels are accessible, labels for newly generated answers given testing samples are not available in real-time. This absence necessitates reliance on indirect measures, such as external knowledge bases, uncertainty quantification and so on, to infer the correctness of generated responses.
> > >
> > > Regarding the i.i.d assumption, as we mentioned before, many foundational machine learning theories operates under the iid assumption to ensure the validity of their theoretical results. Generally speaking, while there're some cases that can go beyond i.i.d, these are often limited to specific scenarios or require additional assumptions or mechanisms. Nonetheless, recognizing the practical limitations of the iid assumption, we have extended our theoretical framework to accommodate distribution shifts, thereby enhancing the generalizability of our approach beyond strictly iid scenarios.
> > >
> > > Besides, we have revised our paper to include all the additional experiments you have raised and refined parts of the main text to eliminate potential confusion and address your concerns. As the discussion deadline approaches, we will appreciate it if you could give us examples that you think "the claims need to be made more formal and informative" and the specific "positioning of the paper" that needs revision.
> > >
> > > Thank you again.

---

### Author Response · Authors · 2024-11-21
**General Response**

Dear Reviewers,

We sincerely thank the reviewers for their time, insightful reviews and constructive suggestions. Overall, it is heartening to note that most of the reviewers found our work to be well motivated(roCe, sEhD), experimental solid (roCe, sEhD) and well-written(roCe, sEhD). To clarify some potential misunderstandings of our paper, we first address some shared concerns of reviewers:

- **FactTest's novelty:** We appreciate the recognition of the novelty in our approach. Unlike traditional conformal prediction (including PAC-style) and conformal language modeling, which focus on **generating prediction sets** that contain the true outcome and provide **coverage guarantees**, FactTest is specifically designed to **identify and filter out incorrect responses** from large language models (LLMs) while **providing Type I error guarantees**. Additionally, while power analysis for conformal prediction primarily concentrates on **the sizes of prediction sets**, our framework emphasizes **the study of Type II errors**. The novelty of FactTest can be summarized as follows:
    - We are **the first** to formulate hallucination detection as a hypothesis testing framework to enforce an upper bound of Type I errors at user-specified significance levels in a finite-sample and distribution-free manner.
    - Motivated by Neyman-Pearson classification, instead of constructing prediction sets for the correct answers using conformal prediction, we propose an one-sided hypothesis testing for the incorrectness of answers.
    - Unlike conformal prediction where all samples are used equally, we prioritize detecting incorrect answers and **utilize only incorrect samples in the calibration data**.
    - We also provide detailed analysis for Type II error control and **derive the optimal score function**.
- **Additional experiments:** According to reviewer Fj97, we have included additional experiments using **more base models** (e.g., Mistral, Llama3.1-Instruct, Tulu2), **more closed-source models** (e.g., Claude, Gemini, GPT-4o), **answer rate analysis**, **more score functions** (KLE of 5 and 10-generation variants), **more baselines** (e.g., SelfCheckGPT) and more Type II error analysis. Notably, any uncertainty quantification method for hallucination detection could be integrated in our framework to serve as the score function and provide correctness guarantees. Though Reviewer roCe raised that we should include PAC conformal prediction as our baseline, we should clarify that **the goal of PAC conformal prediction is different from our factuality testing**, which is not a feasible baseline to compare with.
- **Paper clarity and related works:** Thanks to the suggestions of reviewer Fj97 and sEhD, we have corrected the typos, and revised our writings regarding hallucination, correctness and certainty in our main text. Besides, we have added more prior works about calibration of confidence scores in LLMs and conformal prediction in our updated related works.

---

### Meta-Review · Area_Chair_RAU6 · 2024-12-19

**Metareview:**

This paper proposes a novel strategy for providing statistical guarantees on LLMs for question answering that leverages statistical hypothesis testing techniques to provide guarantees of the form "if the LLM answers the question, then it is correct with high probability". One of the key issues with the paper is its novelty; it is very closely related to the PAC-style conformal prediction literature. While the authors have improved the discussion of the connection in their paper, significant concerns remain.

One specific point of contention is the distinction between constructing prediction sets (the goal in conformal prediction) vs. abstaining from answering the question (the authors' goal). While the authors argue that they are different, their difference is overstated. In particular, the latter problem can be cast in the conformal prediction framework by instead considering a prediction set around a binary classification model designed to predict whether the LLM's answer is correct; then, the LLM answer is only provided if this model outputs {1} (instead of either {0} or {0,1}). There are two important caveats. First, the authors' guarantees are conditioned on the class label (y=0 or y=1); however, class-conditional variants of conformal prediction already exist (and are straightforward modifications; just construct the prediction set for each class separately). Second, the conformal prediction guarantee is slightly stronger than necessary (since it also provides guarantees for when the LLM answer is definitely wrong, which is irrelevant for the authors' problem). Thus, I expect the authors' approach to somewhat outperform the naive application of conformal prediction that I outlined above.

Overall, I agree with the reviewers that a more rigorous comparison to conformal prediction (both theoretically and empirically) is important. I believe re-positioning the paper to account for this connection would significantly strengthen the submission.

Finally, the authors might also consider discussing the following paper leveraging conformal prediction for question answering:

Shuo Li, Sangdon Park, Insup Lee, Osbert Bastani. TRAQ: Trustworthy Retrieval Augmented Question Answering via Conformal Prediction. In NAACL, 2024.

Like the other papers shared on conformal prediction, its focus is on prediction sets rather than abstention, but it is the closest related work that comes to mind.

**Additional Comments On Reviewer Discussion:**

There was significant discussion during the rebuttal period, and while some of the reviewers' concerns were addressed, some broader concerns remain.

---

### Decision · Program_Chairs · 2025-01-22

Reject